# Schistosome infection promotes osteoclast-mediated bone loss

Wei Li[1,2], Chuan Wei[2], Lei Xu[2], Beibei Yu[2], Ying Chen[2], Di Lu[2], Lina Zhang[2], Xian Song[2], Liyang Dong[2], Sha Zhou[2], Zhipeng Xu[2], Jifeng Zhu[2], Xiaojun Chen[2]*, Chuan Su[2]*

1 Department of Clinical Laboratory, Nanjing First Hospital, Nanjing Medical University, Nanjing, P. R. China,
2 State Key Lab of Reproductive Medicine, Jiangsu Key Laboratory of Pathogen Biology, Department of Pathogen Biology and Immunology, Center for Global Health, Nanjing Medical University, Nanjing, P. R. China

* chenxiaojun@njmu.edu.cn (XC); chuansu@njmu.edu.cn (CS)

**Data Availability Statement:** All relevant data are within the manuscript and its Supporting Information files.

## Abstract

Infection with schistosome results in immunological changes that might influence the skeletal system by inducing immunological states affecting bone metabolism. We investigated the relationships between chronic schistosome infection and bone metabolism by using a mouse model of chronic schistosomiasis, affecting millions of humans worldwide. Results showed that schistosome infection resulted in aberrant osteoclast-mediated bone loss, which was accompanied with an increased level of receptor activator of nuclear factor-κB (NF-κB) Ligand (RANKL) and decreased level of osteoprotegerin (OPG). The blockade of RANKL by the anti-RANKL antibody could prevent bone loss in the context of schistosome infection. Meanwhile, both B cells and CD4+ T cells, particularly follicular helper T (Tfh) cell subset, were the important cellular sources of RANKL during schistosome infection. These results highlight the risk of bone loss in schistosome-infected patients and the potential benefit of coupling bone therapy with anti-schistosome treatment.

## Author summary

Schistosomiasis remains an important public health problem in many countries in tropical and subtropical regions, which affects about 200 million people worldwide, with another 700 million considered at risk of infection. Although the primary cause of pathogenesis of schistosomiasis is the granulomatous inflammatory responses, schistosomiasis patients experience long-term hidden pathologies that remain poorly investigated. Here, we found that schistosome infection resulted in RANKL-associated bone loss. Furthermore, our results indicated that both B cells and CD4+ T cells, particularly Tfh cell subset, in the peripheral lymphoid tissues are likely to be the important contributors to bone loss through releasing soluble RANKL. In addition, Tfh cells played a sufficient but not necessary role in schistosome infection-induced bone loss. Our findings highlight the risk of bone loss in schistosome-infected patients and the potential benefit of coupling bone therapy with anti-schistosome treatment.

**Funding:** This work was supported by the grants from the National Key R&D Program of China (No. 2018YFA0507302) and National Natural Science Foundation of China (No. 81871675 and No. 81430052) to CS, the grants from National Natural Science Foundation of China (No. 81871676) and the Natural Science Foundation of Jiangsu Province (No. BK20190082) to XC, and the grant from Nanjing Medical University (No. NMUB2018323) to WL. The funders had no role in study design, data collection and analysis, decision to publish, or preparation of the manuscript.

**Competing interests:** The authors have declared that no competing interests exist.

## Introduction

Schistosomiasis is a chronic helminthic disease that affects more than 200 million people worldwide, with another 700 million considered at risk of infection tab [1,2]. Although the primary cause of pathogenesis of schistosomiasis is the granulomatous inflammatory responses in the liver and intestines [3,4], there is evidence that schistosomiasis patients experience long-term hidden pathologies that have remained poorly characterized. For instance, physical growth impairment in children is linked to parasitic infectious diseases, regardless of nutritional status [5,6]. Indeed, schistosome infection has been associated with growth inhibition in children, even with low parasitic burdens [7], indicating that schistosome infection may influence the skeletal system. Despite the importance of bone tissue in health and development, little is known about the underlying mechanisms of how schistosome infection disrupts bone homeostasis.

Schistosome infection elicits the robust multi-type immune responses beyond the Th2-type response [8], including B-cell response, regulatory T (Treg) cell response, Th17-cell response, and T follicular helper (Tfh) cell response [9–11]. Strikingly, substantial evidence support that schistosome-induced immunity has important impacts on the progression and outcome of multiple other diseases, such as type 1 diabetes and experimental allergic encephalomyelitis [12–14]. Bone is a dynamic tissue that is constantly remodeled by bone-resorbing osteoclasts (OCs) and bone-forming osteoblasts (OBs) in a tightly regulated process [15]. The differentiation of OC precursors into OCs is under the influence of the key osteoclastogenic cytokine receptor activator of nuclear factor-κB ligand (RANKL), which is moderated by osteoprotegerin (OPG), a RANKL's physiological decoy receptor [16]. In fact, an increase in the ratio of RANKL to OPG accelerates the rate of osteoclastic bone resorption in various infectious diseases such as HIV infection and bacterial infection-induced periodontitis [17,18]. Many cell types are capable of producing RANKL, but T cells and B cells are recognized as predominant sources of RANKL in some infectious diseases [18,19]. Given the pivotal roles of T cells and B cells in orchestrating bone metabolism [20,21], it raises the possibility that schistosome infection might influence the skeletal system by altering the host's immune responses, particularly T-cell and/or B-cell responses.

Here, we found that the increased RANKL was associated with schistosome infection-induced bone loss, and showed that both B cells and CD4[+] T cells, particularly Tfh cell subset, were the important contributors to an increased level of RANKL during schistosome infection. Thus, schistosome infection leads to bone loss that reflects the immunological consequences of infection in the host, suggesting that patients with schistosomiasis should pay more attention to bone problems caused by schistosome infection.

## Results

### Schistosome infection results in bone loss in mice

First, we investigated whether schistosome infection had an impact on bone homeostasis. X-radiographic analysis of the femurs showed that mice in the chronic phase of the infection (since 11–13 weeks post-infection) had severe osteoporosis accompanied by a marked decrease in trabecular bone mineral density (Fig 1A). Micro-computed tomography (μCT) analysis of distal femurs showed that schistosome-infected mice had significantly reduced trabecular bone volume and number (Fig 1B). Bone morphometric analysis of distal femurs by μCT showed that trabecular and cortical bone mineral density (BMD, Fig 1C), bone volume (BV), bone volume fraction (BV/TV), trabecular thickness (Tb.Th), number (Tb.N.), and connectivity density (Conn.D., Fig 1D), as well as cortical bone thickness (Ct.Th), area (Ct.Ar), and cortical bone fraction (Ct.Ar/Tt.Ar, Fig 1E) were significantly decreased in schistosome-infected mice. However, no significant difference in tissue volume or total cross-sectional area (TV or

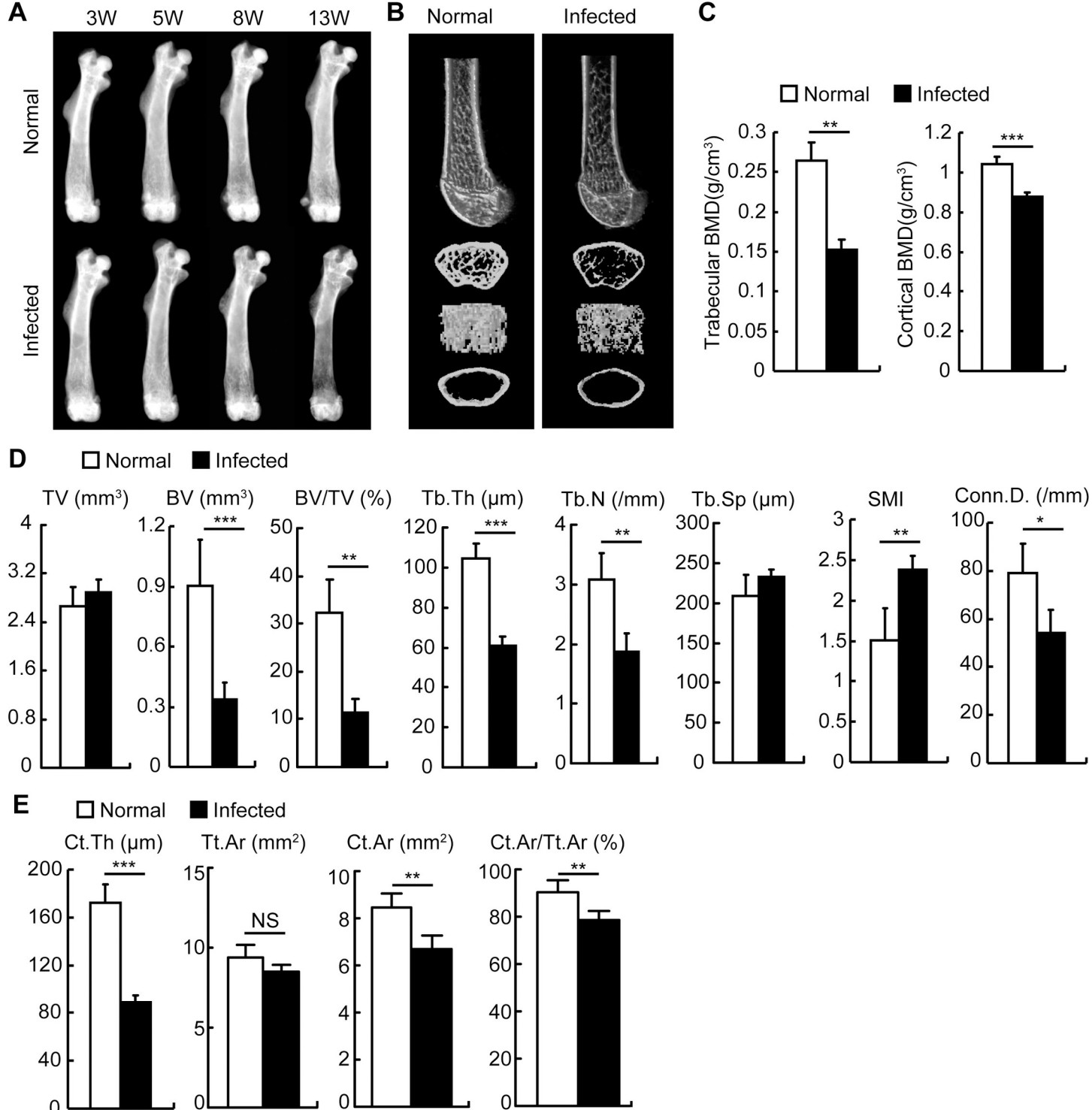

**Fig 1. *S. japonicum* infection drove bone loss in mice.** Male C57BL/6 mice were infected with 12 cercariae of *S. japonicum* per mouse. (A) Mice were sacrificed at 3, 5, 8 or 13 weeks post-infection. Femurs were isolated from male age-matched normal and *S.japonicum*-infected mice, and analyzed by X-ray. Representative of X-ray images of femurs; (B-E) Femurs from mice 13 weeks after infection and age-matched normal mice were analyzed by microcomputed tomography (μCT) scan. (B) Representative μCT images of distal femurs (top, longitudinal view; second, axial view of metaphyseal region; third, 3D view of metaphyseal region; bottom, axial 3D view of the cortical region); (C) Quantitative assessment of trabecular and cortical bone mineral density (BMD); (D) Average quantifications of tissue volume (TV), bone volume (BV), bone volume fraction (BV/TV), trabecular thickness (Tb.Th), number (Tb.N.), space (Tb.Sp.), the structure model index (SMI), and connectivity density (Conn.D.); (E) Average quantifications of cortical bone thickness (Ct.Th), total cross-sectional area (Tt.Ar), bone area (Ct.Ar), and cortical bone fraction (Ct.Ar/Tt.Ar).

Data are representative of three independent experiments, and expressed as the mean ± SD of 6 mice from each group. *, P<0.05; **, P<0.01; ***, P<0.001, NS indicating not significant.

Tt.Ar) or trabecular space (Tb.Sp) was detected, while bone architecture was dramatically altered reflected by an elevated structure model index (SMI) in schistosome-infected mice (Fig 1D and 1E). Indeed, schistosome infection-induced bone loss was also observed in other mouse strains such as BALB/c and ICR (S1 Fig) and patients with schistosomiasis (S1 Table and S1 Text). These results demonstrated that schistosome infection leads to bone loss.

## Aberrant bone resorption causes bone loss in *S.japonicum*-infected mice

*S.japonicum*-infected mice displayed higher bone resorption compared to normal mice, as demonstrated by higher serum levels of the bone resorption marker CTx (C-terminal telopeptide of collagen; Fig 2A), comparable levels of the bone formation marker osteocalcin (Fig 2A), more osteoclasts, and larger osteoclast-covered surface (Fig 2B and 2C), as well as diminished trabecular structure (Fig 2D and 2E). These data suggested that *S.japonicum*-infected mice have reduced bone mass as a consequence of increased osteoclastic bone resorption.

## The increased RANKL is associated with schistosome infection-induced bone loss *in vivo*

Our aforementioned results showed the increases in bone resorption and osteoclast-covered surface in *S.japonicum*-infected mice relative to normal mice (Fig 2A–2C). We next wondered whether the aberrant osteoclast generation was involved in the increased bone resorption during schistosome infection. Thus, we firstly analyzed the absolute number of OCPs (CD3⁻B220⁻NK1.1⁻CD11b$^{-/low}$CD115$^+$CD117$^+$ cells) [22] in bone marrow (BM) of normal and infected mice. Results in Fig 3A showed a comparable number of OCPs in normal and *S.japonicum*-infected mice. Next, we evaluated the levels of osteoclastogenic cytokine RANKL in BM, and found a significant increase in RANKL level in *S.japonicum*-infected mice (Fig 3B). Furthermore, the percentages of RANKL$^+$ cells were substantially increased in bone marrow (S2, S3A and S3B Figs), mesenteric lymph nodes (Fig 3C), and spleens (Fig 3D) during infection. Notably, these effects were also associated with lower OPG expression, resulting in a higher ratio of RANKL to OPG expression in BM (Fig 3B). X-radiographic analysis of the femurs showed an increased trabecular bone mineral density in *S. japonicum*-infected mice treated with anti-RANKL antibody, compared with that in isotype treated-infected mice (Fig 3E). Furthermore, μCT analysis showed that BV, BV/TV (Fig 3F and 3G), the trabecular and cortical BMD (S4A Fig), Tb.N (S4B Fig), Ct. Th, Cr.Ar, Ct.Ar/Tt.Ar (S4C Fig) were significantly increased in schistosome-infected mice treated with anti-RANKL blocking antibody, compared with those in isotype treated-infected mice. In addition, Tb. Th and Conn. D followed the same trend but they did not reach statistical significance (S4B Fig). In contrast, a significant decreased SMI was found in schistosome-infected mice treated with anti-RANKL antibody, while no demonstrable difference in Tb.Sp or Ct.Ar was detected (S4B and S4C Fig). Overall, these results indicated that RANKL blockade ameliorated schistosome infection-induced bone loss in mice.

## Both B cells and CD4$^+$ T cells contribute to the increased level of RANKL during schistosome infection

Given that CD4$^+$ T cells and B cells are viewed as the predominant cellular sources of RANKL in some inflammatory conditions [18,19], we evaluated the distribution of RANKL$^+$ cells in

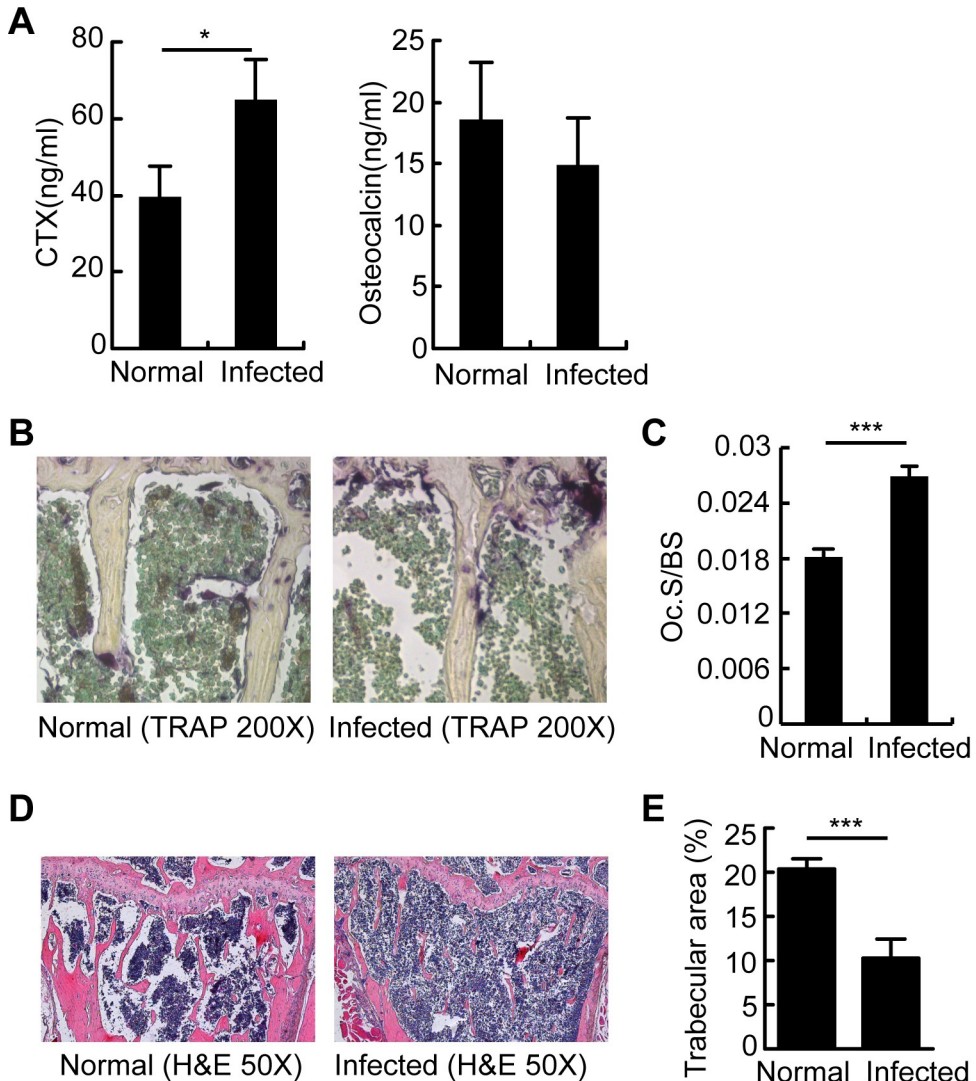

**Fig 2. Aberrant osteoclastogenesis resulted in bone loss in chronic *S.japonicum*-infected mice.** OCs and bone resorption in mice 13 weeks after schistosome infection. (A) C-terminal telopeptide (CTx) and osteocalcin were quantified in the serum of normal and *S.japonicum*-infected mice; (B) Histology and histomorphometry for OCs (purple) were performed in tibias. Representative TRAP-stained tibia sections are shown for normal and *S.japonicum*-infected mice at magnifications of 200×; (C) OC surface per bone surface area (Oc.S/BS) was calculated according to TRAP-stained tibia sections for normal and *S.japonicum*-infected mice; (D) H&E-stained tibias for normal and *S. japonicum*-infected mice at 50× magnification; (E) Trabecular areas were measured using Image-Pro Plus software. Values are given as mean ± SD of 6 mice from each group. Data are representative of three independent experiments. *, P<0.05; ***, P<0.001, NS indicating not significant.

mesenteric lymph nodes and spleens during infection. We found that both CD4+ T cells and B cells were shown to be major *in vivo* sources of RANKL during schistosome infection, and accounted for more than 90% of total RANKL+ cells (Fig 4A and 4B). We next analyzed the relative contribution of CD4+ T cells and B cells to the increased RANKL+ cells in schistosome-infected mice. Results showed that both CD4+ T cells and B cells were likely to be the major contributors to the increased RANKL+ cells during schistosome infection (Fig 4B–4J). A similar increased frequency of RANKL+ lymphocytes was observed in BM in schistosome-infected mice, most of which are constituted by T cells (S3C Fig). Our results, in conjunction with our

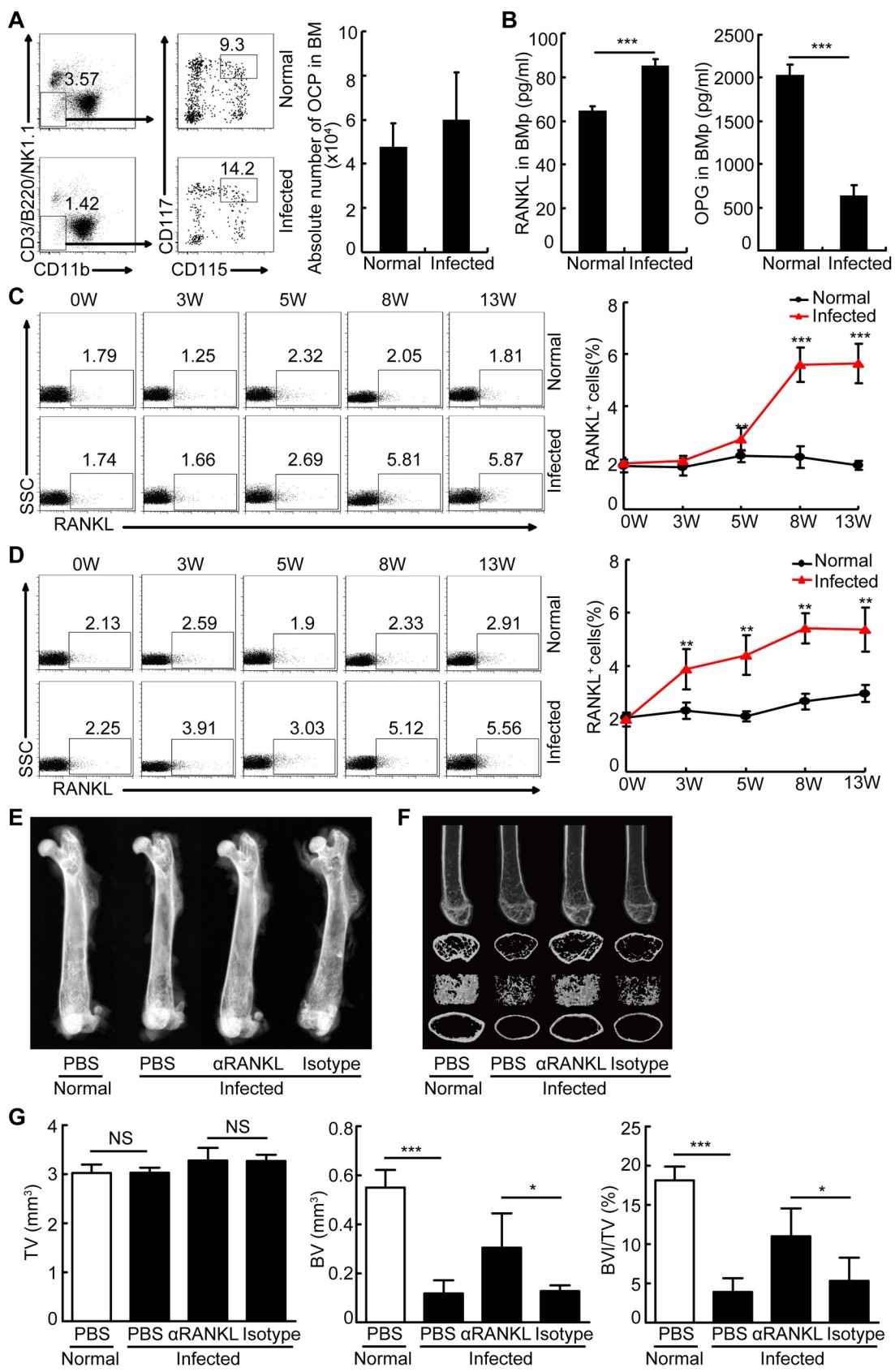

**Fig 3. RANKL blockade ameliorated schistosome infection-induced bone loss in mice.** (A-B) Mouse BMs from mice 8 weeks infected with *S. japonicum* and normal mice were harvested and the cells were surface stained with CD3-PerCP-Cy5.5, B220-PerCP-Cy5.5, NK1.1-PerCP-Cy5.5, CD11b-FITC, CD115-PE, and CD117-APC and analyzed by flow cytometry. Representative flow cytometry data plots and statistics show the percentage and absolute number of OCPs in BM. Data are representative of three experiments with 6 mice in each group; (B) The concentrations of RANKL and OPG in the BM plasma (BMp) of normal mice and mice 8 weeks after infection were quantified by ELISA. Data are representative of three experiments with 6 mice in each group; (C,D) Mouse mesenteric lymph nodes and spleens from mice at 0, 3, 5, 8, or 13 weeks post-infection or age-matched normal mice were harvested and the cells were surface stained with RANKL-PE and analyzed by flow cytometry. Representative flow cytometry data plots and statistics show the frequencies and absolute numbers of RANKL+ cells in mesenteric lymph nodes (C) and spleens (D). Data are representative of three independent experiments with 6 mice in each group; (E) Femurs were isolated from male age-matched normal and *S.japonicum*-infected mice treated with or without anti-RANKL blocking antibody, and analyzed by X-ray. Representative of X-ray images of femurs. (F) Representative μCT images of distal femurs (top, longitudinal view; second, axial view of metaphyseal region; third, 3D view of metaphyseal region; bottom, axial 3D view of the cortical region); (G) Quantitative assessment of tissue volume (TV), bone volume (BV), bone volume fraction (BV/TV) (n = 4–5, pool of two independent experiments). *, $P<0.05$, **, $P<0.01$, ***, $P<0.001$, NS indicating not significant.

observation that there was an evaluated level of osteoclastogenic cytokine RANKL in BM, suggest that CD4+ T cells are likely to be an important contributor to RANKL-mediated bone loss in schistosome-infected mice through releasing soluble RANKL.

## Tfh cells are an important cellular source of RANKL during schistosome infection

CD4+ T cells can differentiate into several subsets including Th1, Th2, Th17, Tfh, and Treg cells in humans and mice during schistosome infection [9]. To investigate the exact cellular mechanisms responsible for schistosome infection-induced bone loss, we analyzed the distribution of RANKL+ CD4+ T cells in mice 8 and 13 weeks after infection, and found that Tfh cells were a major cellular source of RANKL during schistosome infection (Fig 5A and 5B). More importantly, the percentage of RANKL+ Tfh cells within CD4+ T cells had been increased by approximately 5-fold in mice since 8 weeks after infection (Fig 5C), suggesting that Tfh cells were an important contributor to the increased RANKL+ cells *in vivo*, and involved in osteoporosis-associated bone loss during schistosome infection.

Next, we used ICOSL knockout (KO) mice as Tfh cell deficiency model (Fig 5D) [23,24] to address the exact contribution of Tfh cells in the induction of bone loss in *S. japonicum*-infected mice. As expected, Tfh cell deficiency led to a significantly reduced frequency of RANKL-producing cells in CD4+ T cells (Fig 5E), which nicely supported our previous observation that Tfh cells were a major subset of CD4+ T cells producing RANKL during schistosome infection. While the reduction of Tfh cells was only slightly decreased the percentages of total RANKL-producing cells in total lymphocytes (Fig 5F). Indeed, X-radiographic and μCT analysis of the femurs showed that Tfh cell deficiency alone was difficult to reverse schistosome infection-induced bone loss *in vivo* (Figs 5G–5I and S5), which may suggest a possible compensation from multiple sources of RANKL. Collectively, our results indicate that Tfh cells play an important role of contributor, which may be sufficient but not necessary, in schistosome infection-induced bone loss.

## Discussion

The long-term hidden pathological consequences of schistosome infection are poorly defined. Schistosome infection could affect the outcome of many other diseases by altering their host's immunological states [12–14]. In particular, the possible relationship between growth stunting and schistosome infection in humans suggests that schistosome infection may has a negative effect on bone homeostasis [7]. However, there is limited knowledge on how schistosome infection affects bone homeostasis. Given that skeletal homeostasis is dynamically influenced

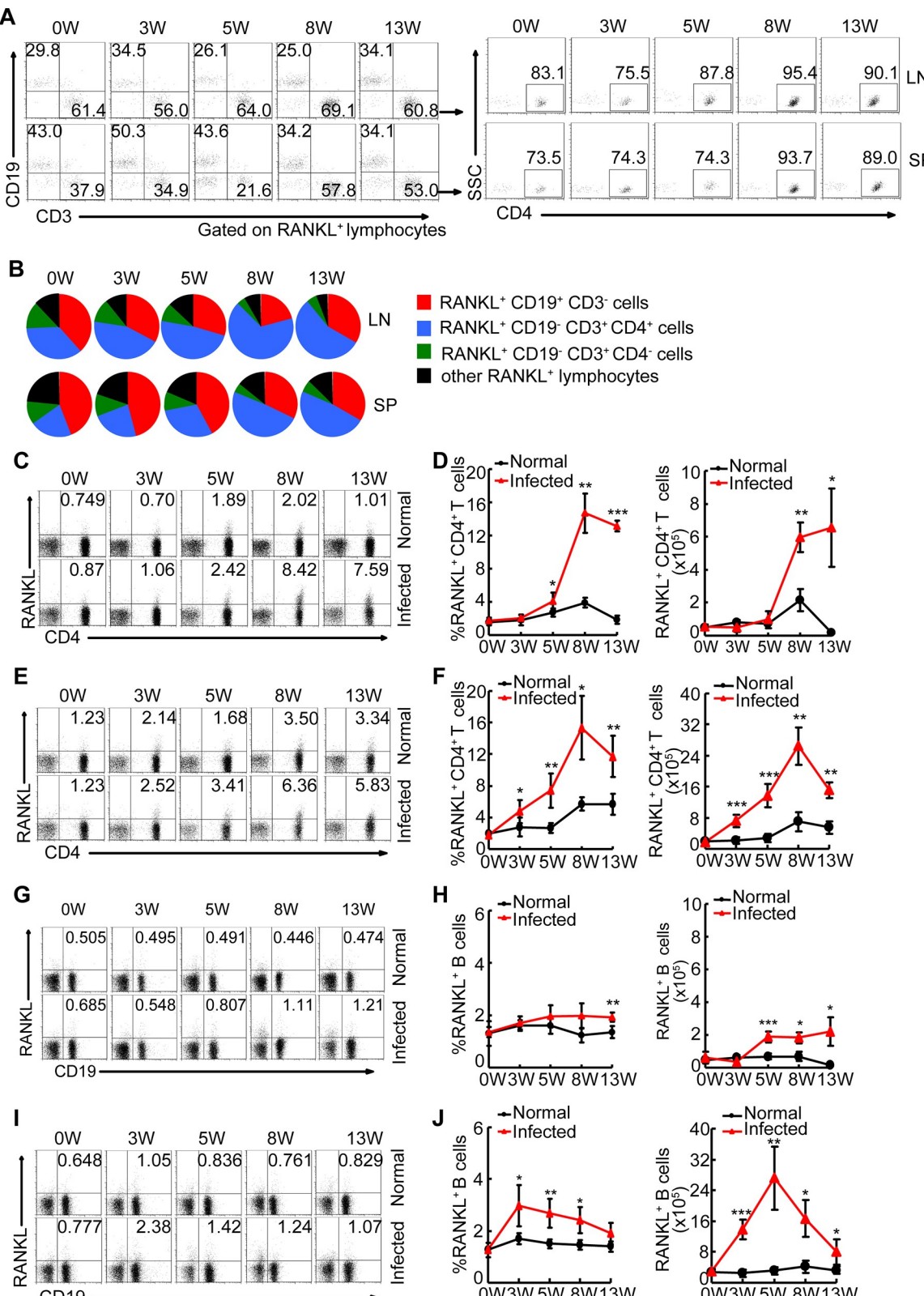

**Fig 4. CD4[+] T cells contributed to the increased level of RANKL during schistosome infection.** Mouse mesenteric lymph nodes and spleens from mice at 0, 3, 5, 8, or 13 weeks post-infection or age-matched normal mice were harvested and the cells were surface stained with CD3- PerCP-Cy5.5, CD4-APC, CD19-FITC, and RANKL-PE, and analyzed by flow cytometry. (A, B) Representative flow

cytometry data plots and statistics show the distribution of B cells (red), CD4[+] T (blue), CD4[-] T (green), and the other cells (black) within total RANKL[+] cells in infected and normal mice. Areas represent the means of percentages of B cells, CD4[+] T, CD4[-] T, and the other cells within total RANKL[+] cells. Gated to RANKL[+] cells; (C-F) Flow cytometry data plots and statistics show the frequencies and absolute numbers of RANKL[+] cells in mesenteric lymph nodes (C-D) and spleens (E-F), Gated to CD3[+] cells; The percentages of RANKL[+] cells of CD4[+] T cells in the graphs were calculated by RANKL[+]CD4[+] / (RANKL[+]CD4[+] + RANKL[-]CD4[+]) in the plots. (G-J) Flow cytometry data plots and statistics show the frequencies and absolute numbers of RANKL[+] cells with CD19[+] cells mesenteric lymph nodes (G-H) and spleens (I-J), Gated to total lymphocytes. Data are representative of three independent experiments with 6 mice in each group. *, P<0.05, **, P<0.01, ***, P<0.001.

by the immune system [20,21,25], it will be of interest to examine the role of schistosome-induced immune responses in bone metabolism in the host. In this study, we showed that the increased RANKL, mainly produced by T cells and B cells, was associated with schistosome-induced bone loss, suggesting that schistosome-infected patients are at higher risk for bone loss and potentially benefit from coupling bone therapy with anti-schistosome treatment.

Accumulating evidence has revealed that some infectious diseases caused by bacteria, viruses, or protozoan parasites, such as *Staphylococcus aureus*, human immunodeficiency virus, or plasmodium, can lead to aberrant bone remodeling that is commonly associated with bone diseases such as osteomyelitis, osteoporosis, or even pathologic fractures [18,26–29]. Although some papers showed that *S. japonicum* or its products suppressed bone resorption in the context of collagen-induced arthritis (CIA), actual schistosome infection exacerbated the severity of arthritis when established CIA mice were infected with schistosome [30–32]. Interestingly, schistosome infection before CIA induction reduced the severity of CIA [31]. Here, we found that actual schistosome infection results in osteoclast-mediated bone loss in mice. Bone homeostasis is maintained by the dynamic balance between osteoclastic bone resorption and osteoblastic bone formation [33]. Because no significant difference in the progress of osteoblast-mediated bone formation was observed between normal and infected mice, we reasoned that the osteoclast-mediated bone resorption was likely to be a major contributor to schistosome infection-induced bone loss. Although some studies indicated that IL-4, IL-10, IL-13, and IFN-γ produced by activated T cells in response to infection likely disrupts OC signaling, the RANK-RANKL signaling pathway is central in the differentiation of OCs [34,35]. A key finding of our study is that schistosome-induced bone loss is associated with the alterations in the RANKL/OPG ratio, and ameliorated by blocking the RANKL-RANK pathway. Indeed, the rates of osteoporosis in schistosomiasis patients were considerably higher than those in healthy controls in the non-endemic area. Thus, these results suggest that patients with schistosomiasis, affecting more than 200 million people worldwide [1,2], might potentially benefit from coupling bone therapy with anti-schistosome treatment.

In homeostatic conditions, osteocytes are major sources of RANKL production, contributing to bone remodeling as demonstrated by previous reports [36]. However, in pathological conditions such as rheumatoid arthritis and HIV infection, other cells including T cells and B cells are likely to be major contributors to increased pathogenic RANKL production [37,38]. Schistosomes are parasitic trematodes living in the inferior mesenteric veins that induce marked B cell and CD4[+] T cell responses in the peripheral lymphoid tissues such as the spleen and MLN [4]. Thus, it is reasonable to hypothesize that schistosome infection results in bone loss through triggering B cell and/or T cell responses in the spleen and MLN, which induce the release of the soluble RANKL and/or home to bone marrow. Although we can not rule out the effects of other cells on RANKL-mediated bone loss in this study, our results indicated that both B cells and CD4[+] T cells in the MLN and spleen are likely to be important contributors to RANKL-associated bone loss through releasing soluble RANKL or homing into bone marrow,

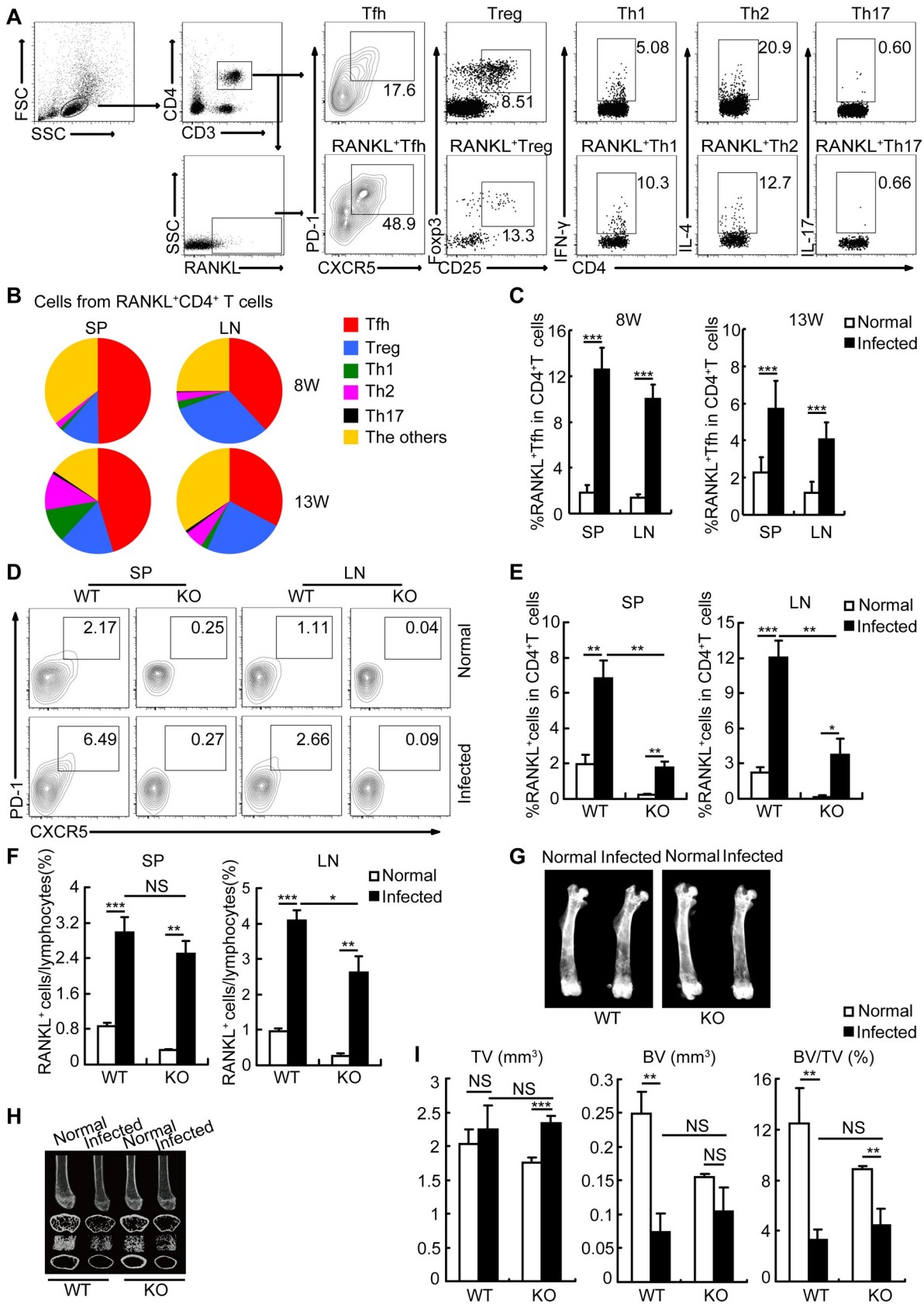

**Fig 5. Tfh cells were identified as a predominant cellular source of RANKL during schistosome infection.** Spleens (SP) and mesenteric lymph nodes (LN) were from wildtype (WT) or ICOSL knockout (KO) mice at 8 or 13 weeks post-infected with or without *Schistosoma japonicum*. (A, B) Representative flow cytometry data plots and statistics show the distribution of Tfh (red), Treg (blue), Th1 (green), Th2 (purple), Th17 (black) and the other CD4+ T cells (orange) within total RANKL+ CD4+ T cells in mice 13 weeks after infection. Areas represent the means of percentages of Tfh, Treg, Th1, Th2, Th17, and the other CD4+ T cells within total RANKL+ CD4+ T cells in infected mice; (C) Flow cytometry data statistics show the frequencies of RANKL+ Tfh cells with CD4+ T cells in LN and SP of normal or infected mice; (D) Representative flow cytometry data plots show the percentages of Tfh cells within CD4+ T cells in SP and LN in WT or ICOSL KO mice infected with or without *Schistosoma japonicum*; (E) Flow cytometry data statistics show the frequencies of RANKL+ cells within CD4+ T cells in LN and SP in WT or ICOSL KO mice infected with or without *Schistosoma japonicum*; (F) Flow cytometry data statistics show the frequencies of RANKL+ cells in LN and SP in WT or ICOSL KO mice infected with or without *Schistosoma japonicum*; (G) Femurs were isolated from WT or ICOSL KO mice infected with or without *Schistosoma japonicum*, and analyzed by X-ray. Representative of X-ray images of femurs; (H) Representative μCT images of distal femurs from WT or ICOSL KO mice infected with or without *Schistosoma japonicum* (top, longitudinal view; second, axial view of metaphyseal region; third, 3D view of metaphyseal region; bottom, axial 3D view of the cortical region); (I) Quantitative assessment of tissue volume (TV), bone volume (BV), bone volume fraction (BV/TV). Data are representative of two independent experiments with 4 mice in each group. *, $P<0.05$, **, $P<0.01$, ***, $P<0.001$, NS indicating not significant.

in which antigen-experienced T and B cells return to specialized niches for the formation of long-lived memory cells [21]. Consistent with this notion, we observed both elevated levels of soluble RANKL and increased frequencies of RANKL+ T cells in bone marrow in schistosome-infected mice. In addition, several previous published literatures also show that RANKL-producing T cells or B cells in the peripheral lymphoid tissues were the dominant drivers of osteoclast-associated bone loss under certain pathological conditions [18,39–41].

Based on the phenotype, cytokine profiles, and functionality, CD4+ T cells are divided into distinctive subsets including type 1 helper T (Th1) cells, Th2 cells, Th17 cells, follicular helper T (Tfh) cells, and regulatory T (Treg) cells [42]. Recent *in vitro* experiments show that Th17 cells can secrete RANKL, and are viewed as an osteoclastogenic helper T cell subset [40]. Consistent with the published literature, our results showed that most Th17 cells produced RANKL in *S.japonicum*-infected mice. Accumulating evidence showed that IL-17 produced by Th17 cells has a role in promoting osteocytes to secrete RANKL [40]. Given that the number of Th17 cells was relatively low in schistosome-infected mice, it is plausible that IL-17-mediated RANKL production may not be a dominant contributor to increased RANKL in schistosome-infected mice. However, the exact contribution of IL-17 to RANKL production during schistosome infection remains to be further investigated. Unexpectedly, here we revealed that Tfh cells were a major *in vivo* source of RANKL. Although Tfh cells deficiency alone reduced significantly the frequency of RANKL-producing cells in CD4+ T cells, it only decreased slightly the percentages of total RANKL-producing cells in total lymphocytes, which suggests that Tfh cells were identified as a predominant subset of RANKL-producing CD4+ T cells, however, other RANKL-producing cells could serve a complementary role.

In conclusion, we demonstrated that the increased RANKL was correlated with the host's osteoclast-mediated bone loss in the context of schistosome infection, and that both B cells and CD4+ T cells, particularly Tfh cells, a dominant subset of RANKL-producing CD4+ T cells, are involved in schistosome infection-induced bone loss *in vivo*. Thus, schistosome infection may be a substantial contributing factor to exacerbated bone loss by altering the host's immune homeostasis. These results highlight the risk of bone loss in schistosomiasis patients and the potential benefit of coupling bone therapy with anti-schistosome treatment.

## Materials and methods

### Ethics statement

Animal experiments were performed in strict accordance with the Regulations for the Administration of Affairs Concerning Experimental Animals (1988.11.1), and all efforts were made to

minimize suffering. All animal procedures were approved by the Institutional Animal Care and Use Committee (IACUC) of Nanjing Medical University for the use of laboratory animals (Permit Number: IACUC-1601213).

## Mice, infection and anti-RANKL blocking antibody treatments

Eight-week-old male C57BL/6J, BALB/c or ICR mice were purchased from the SLAC Laboratory (Shanghai, China). ICOSL$^{-/-}$ C57BL/6J mice were purchased from the Jackson Laboratory (Bar Harbor, ME). Mice were infected percutaneously with 12 *S. japonicum* cercariae (Chinese mainland strain) obtained from infected *Oncomelania hupensis* snails purchased from the Jiangsu Institute of Parasitic Diseases (Wuxi, China). Three, five, eight, and thirteen weeks post-infection (a time point of chronic infection), experiments were performed. In certain experiments, mice 5 weeks after *S. japonicum* infection were administered intraperitoneally with 300 μg/mouse of anti-mouse RANKL blocking antibody (Clone IK22-5; BioXcell, West Lebanon, NH) or isotype control (Clone 2A3; BioXcell) three times per week for 8 weeks.

## X-ray and μCT analysis

The femurs from normal and infected mice were analyzed by a MX-20 X-ray equipment (Faxitron, Tucson, AZ) or a Skyscan 1176 μCT scanner (Bruker, Bremen, Germany). For microcomputed tomography (μCT) analysis, the femurs were removed and kept in 10% neutral-buffered formalin. The parameters of the scan in the 1176 μCT scanner are as follows: Scan duration (6–7 minutes), X-ray voltage (50 kV), X-ray current (455 μA), number of slices (329), Filter (0.5 mm aluminium), Image pixel size (17.76 μm), Camera pixel size (12.59μm), Rotation step (0.6), and Frame averaging (1). Two-dimensional images were used to generate three-dimensional renderings using the CT-Analyser (CTAn; Bruker Micro-CT Software). Both trabecular (metaphyseal) and cortical (metaphyseal-diaphyseal) were selected with reference to the growth plate. The trabecular region commenced about 0.3552 mm (20 image slices) from the growth plate level in the direction of the metaphysis, and extended from this position for a further 1.4208 mm (100 image slices). The cortical region commenced about 2.4864 mm (140 image slices) from the growth plate level in the direction of the metaphysis, and extended from this position for a further 0.5328 mm (170 image slices). According to the published guidelines for the assessment of bone microstructure in rodents using μCT [43], the following indices were calculated with the software which includes trabecular mineral density (BMD), tissue volume (TV), bone volume (BV), bone volume fraction (BV/TV), trabecular thickness (Tb.Th), trabecular number (Tb.N), trabecular space (Tb.Sp), structure model index (SMI, an indicator of the trabecular structures), and connectivity density (Conn.D.), and cortical mineral density (BMD), cortical thickness (Ct.Th), total cross-sectional area (Tt.Ar), cortical bone area (Ct.Ar), and cortical bone fraction (Ct.Ar/Tt.Ar).

## Immunohistochemical analysis

Bones were fixed in 4% formaldehyde and decalcified in 10% EDTA, and 5-μm-thick-paraffin-embedded sections were obtained and stained for TRAP and hematoxylin and eosin (H&E). All images were captured using an Axiovert 200M microscope (Carl Zeiss GmbH, Jena, Germany). Static histomorphometric parameters were measured in sections stained for TRAP using Image-Pro Plus software. The following primary parameters were determined in five fields of view (×200) in one section per mouse: OC surface area and bone surface area. The values obtained from these 5 fields were then averaged and used as the value for that particular animal. From the primary data, the OC surface per bone surface area (Oc.S/BS) was calculated.

## Preparation of BM plasma

BM plasma (BMp) was prepared as described previously with some modifications [44]. Briefly, BMp was prepared by flushing femurs and tibias from one or four mice with 250μL RPMI 1640 containing 0.5% BSA and 10 mM Hepes for OPG or RANKL detection, respectively. Suspended cells were transferred into Eppendorf-type centrifuge tubes, and then cells and debris were removed by repeated (2×) centrifugation at 5,000 g for 10 min.

## ELISA for CTx, osteocalcin, RANKL, and OPG

To quantify the levels of CTx, osteocalcin, RANKL, and OPG in serum or BMp, RATlaps ELISA kit (catalog number AC-06F1, Immunodiagnostic Systems, Fountain Hills, AZ), mouse osteocalcin ELISA kit (catalog number 60–1305, Immutopics, San Clemente, CA), mouse TRANCE/TNFSF11/RANK Ligand ELISA kit (catalog number DY462, R&D Systems, Minneapolis, MN), and mouse osteoprotegerin/TNFRSF11B immunoassay ELISA kit (catalog number MOP00, R&D Systems) with established protocols from the manufacturer were used, respectively. CTx, osteocalcin, RANKL, and OPG were measured in serum or BMp from mice, following an overnight fast.

## Flow cytometry

Single-cell suspensions were prepared by spleens, mesenteric lymph nodes (LN), and BM in PBS containing 1% EDTA followed by red blood cell (RBC) lysis. For surface staining, $1 \times 10^6$ cells per 100 μl were incubated for 30 min at 4˚C with the following fluorescently labeled monoclonal antibodies: CD3-PerCP-Cy5.5 (clone 145-2C11; eBioscience, San Diego, CA), CD4-APC (clone RM4-5; eBioscience), CD4-FITC (clone RM4-5; eBioscience), CD19-FITC (clone MB19-1; eBioscience), CD25-PE-Cy7 (clone PC61.5; eBioscience), B220-PerCP-Cy5.5 (clone RA3-6B2; eBioscience), NK1.1-PerCP-Cy5.5 (clone PK136; eBioscience), CD11b-FITC (clone M1/70; eBioscience), CD115-PE (clone AFS98; BioLegend, San Diego, CA), and CD117-APC (clone 2B8; BioLegend), CXCR5-APC (clone 2G8; BD Biosciences, San Diego, CA), PD-1-PE-Cy7 (clone J43; eBioscience), and RANKL-PE (clone IK22/5; eBioscience). After staining of surface markers, the cells were permeabilized with cold Fix/Perm Buffer, then followed by intracellular staining with IFN-γ-APC (clone XMG1.2; eBioscience), IL-4-APC (clone 11B11; eBioscience), IL-17-APC (clone eBio17B7; eBioscience), or Foxp3-APC (clone FJK-16s; eBioscience) for Th1, Th2, Th17, or Treg cells detection, respectively. The cells were then washed twice in wash buffer before analysis by FACSVerse cytometer (BD Biosciences). FlowJo software (Tree Star, Ashland, USA) was used for the generation of figures and plots. All flow cytometry results were analyzed and plotted using Fluorescence Minus One controls (FMO).

## Statistics

Data were analyzed by unpaired t-test for two-group comparison and ANOVA test for three or more-group comparison. Before analysis, the normality assumption was examined. P-values of less than 0.05 were considered as statistically significant. To account for the type I error inflation due to multiple comparisons, we applied the Bonferroni correction. The analyses were performed using SPSS 11.0 software (IBM), and data are presented as mean ± SD.

## Supporting information

**S1 Fig. *S. japonicum* infection-induced bone loss in BALB/c or ICR mice.** Male BALB/c or ICR mice were infected with 12 cercariae of *S. japonicum* per mouse. Mice were sacrificed at

13 weeks post-infection. Femurs were isolated from male age-matched normal or *S.japonicum*-infected mice, and analyzed by X-ray. Representative of X-ray images of femurs.
(TIF)

**S2 Fig. RANKL was stained and plotted using Fluorescence Minus One controls.** Cells were isolated from the spleen of schistosome-infected mice. (A) Cells were stained with CD3-PerCP-Cy5.5, CD4-APC, and RANKL-PE antibodies. Gated on CD3$^+$ cells. RANKL$^+$CD4$^+$ T cells were analyzed and plotted using Fluorescence Minus One controls (FMO); (B) Cells were stained with CD19-FITC and RANKL-PE antibodies. RANKL$^+$CD19$^+$ B cells were analyzed and plotted using FMO.
(TIF)

**S3 Fig. RANKL$^+$ cells were increased in bone marrow in schistosome-infected mice.** Bone marrow from mice at 13 weeks post-infection or age-matched normal mice was harvested and the cells were surface stained with CD3-PerCP-Cy5.5, CD19-FITC, and RANKL-PE antibodies. (A, B) Representative flow cytometry data plots (A) and statistics (B) show the frequencies of RANKL$^+$ cells in the bone marrow. Data are representative of three independent experiments with 3 mice in each group. $^{**}$, P<0.01; (C) Representative flow cytometry data plots show the distribution of CD19$^+$ B cells, CD3$^+$ T cells, and CD19$^-$CD3$^-$ cells within total RANKL$^+$ cells in bone marrow in infected and normal mice. Gated on RANKL$^+$ cells. Data are representative of three independent experiments with 3 mice in each group.
(TIF)

**S4 Fig. RANKL blockade ameliorated schistosome infection-induced bone loss in mice.** Femurs were isolated from male age-matched normal and *S.japonicum*-infected mice treated with or without anti-RANKL antibody, and analyzed by microcomputed tomography (μCT) scan (n = 4–5, pool of two independent experiments). (A) Quantitative assessment of trabecular and cortical bone mineral density (BMD); (B) Average quantifications of trabecular thickness (Tb.Th), number (Tb.N.), space (Tb.Sp.), the structure model index (SMI), and connectivity density (Conn.D.); (C) Average quantifications of cortical bone thickness (Ct. Th), total cross-sectional area (Tt.Ar), bone area (Ct.Ar), and cortical bone fraction (Ct.Ar/Tt. Ar). n = 4–5, pool of two independent experiments. Data were expressed as the mean ± SD. $^*$, P<0.05; $^{**}$, P<0.01; $^{***}$, P<0.001, NS indicating not significant.
(TIF)

**S5 Fig. Tfh cell deficiency alone failed to reverse schistosome infection-induced bone loss.** Femurs were isolated from wildtype (WT) or ICOSL knockout (KO) mice infected with or without *Schistosoma japonicum*, and analyzed by microcomputed tomography (μCT) scan. (A) Quantitative assessment of trabecular and cortical bone mineral density (BMD); (B) Average quantifications of trabecular thickness (Tb.Th), number (Tb.N.), space (Tb.Sp.), the structure model index (SMI), and connectivity density (Conn.D.); (C) Average quantifications of cortical bone thickness (Ct.Th), total cross-sectional area (Tt.Ar), bone area (Ct.Ar), and cortical bone fraction (Ct.Ar/Tt.Ar). Data are representative of two independent experiments with 4 mice in each group. $^*$, P<0.05, $^{**}$, P<0.01, $^{***}$, P<0.001, NS indicating not significant.
(TIF)

**S1 Table. The rates of bone loss and osteoporosis in schistosomiasis patients and healthy humans.** Schistosomiasis patients (n = 238) from a village in Chizhou City, Anhui province, China. The bone mineral density was detected in schistosomiasis patients by the heel ultrasound. The rates of osteoporosis from the healthy controls in Zhejiang province, a previous schistosomiasis endemic area but announced the elimination of this disease in 1992, were

published in Chinese academic journal [45]. The rates of osteoporosis or bone loss between schistosomiasis patients and healthy humans were analyzed by Pearson Chi-Square Test. (DOCX)

**S1 Text. Supporting text.** This file contains detailed materials and methods for S1 Table. (DOC)

## Acknowledgments

We thank Wen Jiang (Nanjing Medical University) for language editing for the manuscript.

## Author Contributions

**Conceptualization:** Xiaojun Chen, Chuan Su.

**Data curation:** Wei Li, Chuan Wei, Lei Xu, Sha Zhou, Xiaojun Chen.

**Formal analysis:** Chuan Wei, Lei Xu, Xiaojun Chen.

**Funding acquisition:** Wei Li, Xiaojun Chen, Chuan Su.

**Investigation:** Wei Li, Lei Xu, Beibei Yu, Ying Chen, Di Lu, Lina Zhang, Xian Song, Liyang Dong, Sha Zhou, Zhipeng Xu, Jifeng Zhu.

**Methodology:** Wei Li, Chuan Wei, Lei Xu, Beibei Yu, Ying Chen, Di Lu, Lina Zhang, Xian Song, Zhipeng Xu.

**Project administration:** Xiaojun Chen, Chuan Su.

**Resources:** Xiaojun Chen, Chuan Su.

**Software:** Wei Li, Xiaojun Chen.

**Supervision:** Xiaojun Chen, Chuan Su.

**Writing – original draft:** Wei Li, Xiaojun Chen.

**Writing – review & editing:** Xiaojun Chen, Chuan Su.

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
