## [Decision Letter · Decision Letter 0]

14 Jul 2020

Dear Dr. Su,

Thank you very much for submitting your manuscript "Schistosome Infection Promotes Osteoclast-mediated Bone Loss" for consideration at PLOS Pathogens. As with all papers reviewed by the journal, your manuscript was reviewed by members of the editorial board and by several independent reviewers. In light of the reviews (below this email), we would like to invite the resubmission of a significantly-revised version that takes into account the reviewers' comments.

We cannot make any decision about publication until we have seen the revised manuscript and your response to the reviewers' comments. Your revised manuscript is also likely to be sent to reviewers for further evaluation.

Sincerely,

De'Broski R Herbert

Guest Editor

PLOS Pathogens

Edward Pearce

Section Editor

PLOS Pathogens

Kasturi Haldar

Editor-in-Chief

PLOS Pathogens

orcid.org/0000-0001-5065-158X

Michael Malim

Editor-in-Chief

PLOS Pathogens

orcid.org/0000-0002-7699-2064

Reviewer's Responses to Questions

**Part I - Summary**

Reviewer #1: This study investigates, the relationships between chronic schistosome infection and bone metabolism in a mouse model. The data suggest schistosome infection induces bone resorption, increased RANKL by follicular helper T cells and diminished OPG. While the studies and outcomes are very interesting, the data is somewhat preliminary and there are numerous concerns regarding the experimental design and its rigor, that need to be addressed.

Reviewer #2: Schistosomiasis appears to result in immunological changes that impact bone metabolism. To study this, these authors investigated the relationships between chronic schistosome infection and bone metabolism by using a mouse model of chronic schistosomiasis. Schistosome infection resulted in osteoclast-mediated bone loss, and assays demonstrated activation of NF-κB ligand and decreased level of a molecular, osteoprotegerin. The blockade of NF-κB ligand could prevent the bone loss, which was quite an intriguing result. Follicular helper T (Tfh) cells were identified as a predominant cellular source of NF-κB ligand during schistosome infection. These results might point the way to novel therapeutic interventions against bone loss in schistosome-infected patients.

In general, the paper is well written and the conclusions are supported by the data. Appropriate controls and statistical comparisons are provided. One comment is that on page 10 the authors note that many other infections can perturb bone metabolism and growth. This begs the obvious question that the authors should address: namely, how confident can they be that the perturbations are unique to schistosomiasis? Were experiments done with other pathogens to demonstrate that the perturbations in NF-κB ligand and the decreased level of osteoprotegerin were unique to Schistosomiasis or are these noted in other infections?

More comment on the precise scope of the skeletal abnormalities in humans would improve the discussion. Are the extensive abnormalities noted in mice also seen in humans? How big of a public health impact does Schistosomiasis have on human bone disease?

Reviewer #3: (No Response)

**Part II – Major Issues: Key Experiments Required for Acceptance**

Reviewer #1: The therapeutic study where RANKL was ablated in the mice to ameliorate bone loss, is poorly explained and the data consists of a single panel of X-ray data. This is not quantitative data and the images are not informative. Full uCT and turnover markers are needed for this study. Irrespective, it’s unclear what this data means and the conclusions of this study are erroneous, and do not prove as claimed, that aberrant RANKL underlies the mechanism. Blocking RANKL would stall bone resorption irrespective of the mechanism by suppressing basal resorption. At the very least an uninfected control group treated with RANKL antibody needs to be performed for comparison. But even with that, interpretation would be difficult. A T cell conditional RANKL KO experiment would be the most convincing, however B cells also appear to make RANKL in this model (the authors dismiss this) and other source such as BMSC and osteocytes that are responsive to IL-17 from Th17, were not investigated as putative sources. The relevant sources of RANKL remain unclear at this time.

The reporting of methodology and required indices for uCT, does not correspond to the published guidelines (Bouxsein, M. L., et al. (2010). "Guidelines for assessment of bone microstructure in rodents using micro-computed tomography." Journal of Bone and Mineral Research 25(7): 1468-1486). The reported indices and methodology should be revised extensively per these guidelines.

Elevated SMI would not mask the lack of changes in Tb. Sp. as suggested. SMI is a mathematical extrapolation of bone architecture based on a structural model, while Tb. Sp. is an actual measurement. Why Tb. Tp. Does not change is perplexing given both Tb. N. and Tb. Th. Are significantly diminished.

How was cortical and trabecular BMD calculated by uCT? These is not a traditional uCT indices! TV.D (mg HA per cm3) is the recommended uCT parameter.

There are many required cortical indices that are missing and should to be presented, especially given the image seems to suggest significant decline in cortical bone mass.

Analysis of the axial skeleton should also be performed.

Figure 4 shows that both CD4 and CD19 lymphocytes make RANKL during infection, however the authors conclude that only T cells contribute to bone resorption. This is surprising. The data also needs to be normalized for total cell number as % is very misleading. The number of total B cells in the spleen and bone marrow is significantly higher than that of T cells, and hence even a small change in B cell RANKL may considerably outweigh a relatively larger increase in T cell RANKL. More detailed studies would be needed to ascertain the relevant sources of RANKL. Why were CD8 T cells excluded from the study, they are also known to make RANKL.

Th17 T cells are not only producers of RANKL, they are inducer of RANKL from other cells including osteoblast lineage cells, which were not examined.

Figure 5. A full uCT analysis of cortical and trabecular bone and bone turnover markers are needed to substantiate the conclusion that Tfh mediates the bone loss in infected mice. The changes in trabecular BMD (whatever that represents) are less than half of those shown for WT mice. Thus, a partial role for Tfh only may be indicated. That is responsible for the rest of the activity?

Throughout, the X ray images are not quantitative and are not convincing of an effect.

Reviewer #2: No issues.

Reviewer #3: (No Response)

**Part III – Minor Issues: Editorial and Data Presentation Modifications**

Reviewer #1: A lot of grammatical errors. Recommend a native English speaker review the manuscript, or Grammar software beyond Word should be utilized.

Please check spelling of “Infected” in Figure 2A.

It is suggested that anti-schistosomal drug treatment coupled with Denosumab may be beneficial in improving bone health. Any antiresorptive drug would likely be effective in improving bone mass. There does not seem to be a specific indication for use of Denosumab simply because RANKL is increased. As schistosome infection occurs largely in resource-limiting settings, Denosumab with its high cost would not seem to be a practical option compared to a bisphosphonate which would have similar effect. Furthermore, if schistosomes are eradicated pharmacologically as suggested, would this not alone prevent further bone loss? In which case Denosumab would likely no longer be necessary.

Page 11 line 17: however, no evidence shows that schistosomes or schistosome-derived antigens have demonstrable effect on osteocytes. The meaning or relevance of this statement is unclear.

Page 13, line 18: The contention that reference 42 documents RANKL expression by Tregs is not valid. In fact, this interesting study shows that a sub-set of Tregs may transdifferentiate into Th17 cells that secrete RANKL. The Tregs themselves are not generally considered a source of RANKL, and if anything, are anti-osteoclastogenic.

Immunohistochemical studies/ How was the osteoclast quantifications performed? No methodology is presented

Flow methods. No mention of FMO and isotype controls for gating.

Reviewer #2: No concerns.

Reviewer #3: (No Response)

PLOS authors have the option to publish the peer review history of their article (what does this mean?). If published, this will include your full peer review and any attached files.

Reviewer #1: No

Reviewer #2: No

Reviewer #3: No
---

## [Decision Letter · Decision Letter 1]

27 Jan 2021

Dear Dr. Su,

Thank you very much for submitting your manuscript "Schistosome Infection Promotes Osteoclast-mediated Bone Loss" for consideration at PLOS Pathogens. As with all papers reviewed by the journal, your manuscript was reviewed by members of the editorial board and by several independent reviewers. The reviewers appreciated the attention to an important topic. Based on the reviews, we are likely to accept this manuscript for publication, providing that you modify the manuscript according to the review recommendations.

Minor edits need to be made.

Sincerely,

De'Broski R Herbert

Guest Editor

PLOS Pathogens

Edward Pearce

Section Editor

PLOS Pathogens

Kasturi Haldar

Editor-in-Chief

PLOS Pathogens

orcid.org/0000-0001-5065-158X

Michael Malim

Editor-in-Chief

PLOS Pathogens

orcid.org/0000-0002-7699-2064

Minor edits need to be made.

Reviewer Comments (if any, and for reference):

Reviewer's Responses to Questions

**Part I - Summary**

Reviewer #1: The authors have addressed in part my concerns, and the manuscript is improved. However, there are still some relatively simple to fix issues that were not effectively addressed and need to be thought through more carefully and fixed. The interest and novelty of the study remains a strength.

Reviewer #3: I am satisfied with the modifications and responses and changes made by the authors in response to the reviewers' comments. They have addressed my concerns satisfactorily.

**Part II – Major Issues: Key Experiments Required for Acceptance**

Reviewer #1: N/A

Reviewer #3: None

**Part III – Minor Issues: Editorial and Data Presentation Modifications**

Reviewer #1: 1. The authors have failed to address the previous concerns regarding the role of B cell RANKL production in this model. B cells are still being dismissed based on the fact that a higher percentage of T cells (60%) express RANKL. This is an incorrect comparison because, as previously explained, the total number of B cells are dramatically higher than CD4 T cells, especially in the bone marrow where bone resorption occurs and where B cell subpopulations may represent up to 25% of the total cell population. By comparison CD4 T cells normally represent only 2-3% of total bone marrow. Similar, in spleen, B cells are 50% of total cells while CD4 represent only about 7%. Consequently, even if only half as many B cells are making RANKL (30% is reported here), because there are 5X as many B cells in total, their contribution may be very significant and outweigh that of T cells. Without further data as to the relative contributions of B and T cells, the authors need to stop claiming that T cells, and in particular CD4 T cells, are the major drivers of bone resorption and make allowance for a role of both T and B cells, and potentially a greater role of B cells given their relatively larger numbers.

2. The authors contend that a RANKL antibody control group is unnecessary based on some previous studies that did not use the appropriate controls. Most of the papers presented in proof, are very dated and/or published in low impact journals and/or were simply poorly reviewed. The important of complete controls, cannot be underemphasized. In this case it is not possible to conclude that increased RANKL is driving the events simply because depleting RANKL ablated the effect. Suppressing osteoclasts by any means (as previously explained) would reduce baseline resorption and prevent bone loss. This is not prima facia evidence of a cause-effect relationship, and consequently, the authors should not over-interpret the data and claim that this experiment proves RANKL involvement. While increased RANKL is likely, all that can be concluded from this experiment is that an anti-resorptive strategy may also prevent bone disease in human patients.

3. uCT Indices. I was not suggesting that you report all of the indices in the Guidelines, only that the authors use the recommended units and document the methodology as indicated. These guidelines were developed and adopted by the American Society for Bone and Mineral Research, so that different studies can be compared directly and parameters reported without confusion. For example, “bone volume over tissue volume (Tb. Vol/TV)” is neither the correct index nor notation. This should be bone volume fraction (BV/TV). Cortical bone volume as a fraction of total bone volume (Co. Vol/TV) Co. is also not the appropriate unit. This should be Cortical area fraction and is abbreviated as Ct.Ar/Tt.Ar. The guidelines state: “The minimal set of variables that should be reported for cortical regions includes total cross-sectional area (Tt.Ar), cortical bone area (Ct.Ar), cortical thickness (Ct.Th), and cortical bone fraction (Ct.Ar/Tt.A)”

4. The non-standard indices generated by your Brucker are very confusing and should be converted to common notation used in the Guidelines. There are many Skyscan users who report data in the expected format so it may necessary to request help from Brucker in converting the indices or units.

5. Furthermore, none of the machine settings are presented (e.g. integration time, tube voltage and current), number of slices and where they were measured. Please upgrade the methods to normal bone specifications.

6. Some new uCt has been added for the RANKL ablation experiment Figure 4G, and in 5I but trabecular BMD presented is again not the key index and BV/TV needs to be shown here instead. The data in the supplemental section for BV/TV looks quite good so no reason not to present this as the primary data. The BMDs presented is a dispensable index in uCT.

Reviewer #3: (No Response)

PLOS authors have the option to publish the peer review history of their article (what does this mean?). If published, this will include your full peer review and any attached files.

Reviewer #1: No

Reviewer #3: No
---

## [Editor Report · Decision Letter 2]

9 Mar 2021

Dear Dr. Su,

We are pleased to inform you that your manuscript 'Schistosome Infection Promotes Osteoclast-mediated Bone Loss' has been provisionally accepted for publication in PLOS Pathogens.

Best regards,

De'Broski R Herbert

Guest Editor

PLOS Pathogens

Edward Pearce

Section Editor

PLOS Pathogens

Kasturi Haldar

Editor-in-Chief

PLOS Pathogens

orcid.org/0000-0001-5065-158X

Michael Malim

Editor-in-Chief

PLOS Pathogens

orcid.org/0000-0002-7699-2064

sufficiently improved manuscript
---

## [Editor Report · Acceptance letter]

16 Mar 2021

Dear Prof. Su,

We are delighted to inform you that your manuscript, "Schistosome Infection Promotes Osteoclast-mediated Bone Loss," has been formally accepted for publication in PLOS Pathogens.

Best regards,

Kasturi Haldar

Editor-in-Chief

PLOS Pathogens

orcid.org/0000-0001-5065-158X

Michael Malim

Editor-in-Chief

PLOS Pathogens

orcid.org/0000-0002-7699-2064